# Regional Temperature-Sensitive Diseases and Attributable Fractions in China

**DOI:** 10.3390/ijerph17010184

**Published:** 2019-12-26

**Authors:** Xuemei Su, Yibin Cheng, Yu Wang, Yue Liu, Na Li, Yonghong Li, Xiaoyuan Yao

**Affiliations:** National Institute of Environmental Health, Chinese Center for Disease Control and Prevention, Beijing 100021, China; suxuemei0918@163.com (X.S.); ybcheng123@163.com (Y.C.); wangyu@nieh.chinacdc.cn (Y.W.); liuyue@nieh.chinacdc.cn (Y.L.); lina@nieh.chinacdc.cn (N.L.)

**Keywords:** extreme temperature, sensitive disease, attributable fraction, multi-region study, regional differences

## Abstract

Few studies have been carried out to systematically screen regional temperature-sensitive diseases. This study was aimed at systematically and comprehensively screening both high- and low-temperature-sensitive diseases by using mortality data from 17 study sites in China located in temperate and subtropical climate zones. The distributed lag nonlinear model (DLNM) was applied to quantify the association between extreme temperature and mortality to screen temperature-sensitive diseases from 18 kinds of diseases of eight disease systems. The attributable fractions (AFs) of sensitive diseases were calculated to assess the mortality burden attributable to high and low temperatures. A total of 1,380,713 records of all-cause deaths were involved. The results indicate that injuries, nervous, circulatory and respiratory diseases are sensitive to heat, with the attributable fraction accounting for 6.5%, 4.2%, 3.9% and 1.85%, respectively. Respiratory and circulatory diseases are sensitive to cold temperature, with the attributable fraction accounting for 13.3% and 11.8%, respectively. Most of the high- and low-temperature-sensitive diseases seem to have higher relative risk in study sites located in subtropical zones than in temperate zones. However, the attributable fractions for mortality of heat-related injuries were higher in temperate zones. The results of this research provide epidemiological evidence of the relative burden of mortality across two climate zones in China.

## 1. Introduction

As global temperatures rise, extreme temperature events are expected to become more intense, more frequent and longer by the end of the twenty-first century [1,2]. Extreme temperature events have a variety of adverse effects upon human health, which have contributed to increased mortality due to various diseases [3,4,5,6], including communicable and non-communicable diseases [7,8]. Many previous multi-city studies have shown that both high and low temperature could increase mortality or morbidity [9,10] and have delayed effects.

For example, a study in the United States [11] investigated 31 common diseases in 1943 counties, and showed that extreme heat could increase the rates of hospital admission for specific diseases, such as influenza, electrolyte disturbance and renal failure. A study in China conducted in 272 cities [12] showed that both high and low temperatures increased mortality, especially from circulatory and respiratory diseases, and the burden of death varied by climatic zone. Similar studies were also conducted in France [13], Japan [14] and Vietnam [15]. However, most of the previous studies mainly focused on common diseases, such as cardiovascular, cerebrovascular, respiratory and infectious diseases. Few studies have paid attention to relatively rare diseases such as injuries and nervous disease [16,17].

Studies that cover a comprehensive spectrum of diseases remain scarce, especially in China. Therefore, our study was aimed at examining the association between extreme temperature and a spectrum of diseases besides circulatory and respiratory diseases, in order to systematically and comprehensively screen both high- and low-temperature-sensitive diseases in temperate and subtropical regions, and to assess the mortality burden of sensitive diseases through a multi-city study in China.

## 2. Materials and Methods

### 2.1. Study Regions

Seventeen cities or counties in 10 of 11 meteorological geographic zones in China (shown in Figure 1) were selected as study sites to screen regional diseases sensitive to extreme temperature and to assess the burden of mortality ascribed to extreme temperature. The study sites were divided into two parts according to climatic zones: the subtropical zone (N = 11) and temperate zone (N = 6).

### 2.2. Mortality Data

Daily mortality data for 17 study sites in China during 2014–2017 were collected from the local Centers for Disease Control and Prevention. In the analysis, the following causes of death, according to the International Classification of Diseases, Tenth Revision (ICD-10), were used: all-cause (A00–Z99), infectious (A00–B99), endocrine (E00–E35), diabetes (E10–E14), nervous system (G00–G99), circulatory system (I00–I99), respiratory system (J00–J99), digestive system (K00–K93), genitourinary system (N00–N39) and injury (S00–T98, V01–V99, W00–X59, X60–Y89). Diseases of the circulatory system were subdivided into cardiovascular disease (I20–I25) and cerebrovascular disease (I60–I69), which included intracerebral hemorrhage (I61), cerebral infarction (I63), stroke not specified as hemorrhage or infarction (I64) and sequelae of cerebrovascular disease (I69). 

Diseases of the respiratory system were subdivided into acute infectious respiratory diseases (J09–J18) and chronic lower respiratory diseases (J40–J47).

### 2.3. Meteorological and Air Pollution Data

Daily meteorological data from the national meteorological station at each study site were provided by the China Meteorological Administration, including daily maximum, mean and minimum temperatures (°C), daily mean barometric pressure (hpa) and daily mean relative humidity (RH, %). Daily air pollution data for each site were provided by local Municipal Ecological Environment Bureaus, including particulate matter of mass median aerodynamic diameters less than 2.5 μm (PM_2.5_; 24 h mean in μg/m^3^) and ozone (O_3_; 8 h mean in μg/m^3^). 

### 2.4. Statistical Analysis

#### 2.4.1. Two-Stage Analysis

In the first stage, a distributed lag nonlinear model (DLNM) with quasi-Poisson regression was applied to quantify the association between extreme temperature and cause-specific mortality at each study site. The DLNM model was as follows:(1)$$\log\left[E\left(Y_{it}\right)\right]=\alpha +\mathrm{cb}\left(T_{it},\mathrm{lag}=30\right)+\mathrm{ns}\left(time_{i},\;\mathrm{df}=7/\mathrm{year}\right)+\mathrm{ns}\left(HR,\;4\right)+\;\mathrm{ns}\left(P,\;4\right)+\mathrm{ns}\left(PM_{2.5},\;5\right)+\mathrm{ns}\left(O_{3},\;5\right)+\beta DOW_{it}$$
where *Y_it_* represents the number of deaths on day *t* at site *i*; *α* is the intercept; *T_it_* is the daily maximum temperature on day *t* at site *i*; the lag day up to 30 reflects the maximum lag day of the temperature effect; cb refers to the cross-basis function, which specifies the exposure–lag–response relationship simultaneously in the exposure–response and lag–response dimensions; a quadratic B spline with two knots was used for temperature, and a natural spline with 5 df for lag; ns represents the natural cubic spline; DOW stands for day of the week, which was entered as a categorical variable; and *β* is the coefficient of DOW [18,19]. The mean relative humidity, mean barometric pressure, PM_2.5_, O_3_, long-term trend time and DOW were considered as potential confounders. A correlation analysis of O_3_, PM_2.5_, temperature and relative humidity was conducted for our 17 study sites. The correlation coefficient was between 0.01 and 0.5, thus multicollinearity was not a major concern [20], and all three pollutants were included in the model. Long-term trends were modeled with seven degrees of freedom *(df*) per year. Mean relative humidity (HR) and mean barometric pressure (P) were modeled as a quadratic n-spline with 4 df. O_3_ and PM_2.5_ were modeled as a quadratic n-spline with 5 df. The minimum mortality temperature (MMT) was used as the optimum and reference temperature [12]. The 2.5th and 97.5th percentiles of daily maximum temperature in the entire year were chosen as the extreme low and high temperatures in the model, respectively. 

In the second stage, we obtained the best linear unbiased prediction (BLUP) of each site and specific cumulative (≤30 lag days) associations between temperature and mortality at both regional (climatic zone) and overall levels by using a recently developed multivariate meta-regression [21]. Cumulative relative risk (CRR) under specific lag days was calculated to evaluate the effects of extreme temperature on mortality, and to screen the diseases sensitive to extreme temperature at the overall and regional level compared to MMT. Diseases were considered as sensitive to extreme temperature if the cumulative RR was greater than 1, and the 95% confidence interval (CI) did not contain 1.

#### 2.4.2. Estimation of Attributable Fractions

Based on the above steps, diseases sensitive to extreme temperature was screened, and then the AF values were calculated to estimate the mortality burden caused by the temperature of these diseases. For each day at each study site, we calculated the cumulative RR for the spectrum of diseases by comparing it to the MMT. The attributable deaths and AFs for the present day and 30 lagged days were then calculated according to a backward perspective [22]. 

We obtained the total counts of deaths attributable to non-optimum temperature by summing the contributions from all days in the series and determined the total AF by dividing the total number of deaths by the total number of attributable deaths. We empirically calculated the AFs associated with cold and heat by summing the subsets of days with relevant temperature ranges according to each region’s specific temperature threshold (that is, minimum daily maximum temperature up to the MMT and MMT up to maximum daily maximum temperature, respectively). Finally, we calculated the empirical confidence interval (eCI) values through Monte Carlo simulations [23] and the related 2.5th and 97.5th percentiles of multivariate normal distribution were interpreted as 95% empirical confidence intervals [23]. 

#### 2.4.3. Assessing Regional Differences

In order to assess the differences in the risk of mortality among people in the two climate zones, we repeated the aforementioned 2-stage analysis by region. Then we performed the significance test on the difference between effect estimates for the two subgroups using the following formula [24]:(2)$$Z=\left(E_{1}-E_{2}\right)/\sqrt{SE{\left(E_{1}\right)}^{2}+SE{\left(E_{2}\right)}^{2}}$$
where *Z* is the *Z*-test, *E*_1_ and *E*_2_ are the effect estimates for two categories (such as temperate zone and subtropical zone), and SE (*E*_1_) and SE (*E*_2_) are their respective standard errors.

### 2.5. Sensitivity Analysis

Sensitivity analysis was used to validate the stability of the model by extracting air pollutants (PM_2.5_, O_3_, CO), mean relative humidity, and mean barometric pressure from the model and by using alternative maximum lags of 7, 14 and 21. We also controlled for the 2-day average concentrations of fine particulate matter and ozone (as an indicator of air pollution) in another analysis. The results are shown in Appendix A.

Data analysis was conducted using R software (version 3.5.1, R Foundation for Statistical Computing Platform 2013). The dlnm package [25] and mvmeta package [26] were used for fitting DLNM and meta-analysis, respectively. For all statistical tests, 2-tailed *p*-value < 0.05 was considered as statistically significant.

### 2.6. Ethics Approval and Consent to Participate

Ethical approval for this study was granted by the National Institute of Environmental Health, Chinese Centers for Disease Control and Prevention.

## 3. Results

### 3.1. Descriptive Statistics

Table 1 shows the descriptive statistics of daily maximum temperature, mean relative humidity, mean barometric pressure and air pollutants at 17 study sites during 2014–2017. The mean daily maximum temperature ranged from 5.8 °C (range: −34.1–41.7 °C) in Hailar to 27.1 °C (range: 6.5–36.9 °C) in Shenzhen. The 2.5th and 97.5th percentiles of daily maximum temperature varied across study sites, and ranged from 32 to 38 °C and from −24 to 14 °C, respectively. Daily mean concentration of PM_2.5_ and O_3_ varied across study sites and ranged from 29.1 μg/m^3^ to 108 μg/m^3^ and from 54 μg/m^3^ to 122 μg/m^3^, respectively.

Table 2 shows a summary of descriptive statistics on an average number of daily cause-specific deaths at 17 study sites in China. During 2014–2017, a total of 1,380,713 all-cause deaths were recorded. There were 535,960 (39%) deaths due to diseases of the circulatory system, among which hypertension, ischemic heart disease, cerebral infarction, intracerebral hemorrhage, stroke and sequelae of cerebrovascular diseases accounted for 5%, 31%, 14%, 17%, 1% and 7%, respectively. There were 199,416 (14%) deaths due to diseases of the respiratory system, among which influenza and pneumonia and chronic obstructive pulmonary disease accounted for 18% and 53%, respectively. Diseases of the digestive system and genitourinary system, endocrine diseases, diseases of the nervous system, infectious diseases and injuries accounted for 2% (30,121), 0.7% (10,311), 2.5% (34,361), 0.9% (12,128), 0.6% (7951) and 6% (88,360) of total mortality, respectively.

### 3.2. Association between Extreme Temperature and Cause-Specific Mortality

We found that extreme temperature increased mortality due to multiple diseases, including diseases of the circulatory system and respiratory system, endocrine diseases, injuries and diseases of the nervous system, compared with MMT. However, CRR varied by region and cause of death. Figure 2 shows the overall exposure–response relationship between daily maximum temperature and cause-specific mortality by 30 d lag at 17 study sites in China. The optimum temperature was slightly different for different causes of death. The optimum temperature for endocrine diseases was the highest at an MMT of 30.6 °C, followed by nervous system diseases (MMT = 26.3 °C), circulatory system diseases (MMT = 23.4 °C), respiratory system diseases (MMT = 23.2 °C) and injuries (MMT = 21.8 °C). The overall exposure–response relationship between daily maximum temperature and digestive, urinary and infectious diseases by 30 d lag at 17 study sites in China were shown in Appendix A.

Table 3 shows that the highest CRR of 1.45 (95% CI: 1.27–1.65) associated with extreme heat was observed for injuries, followed by diseases of the nervous system (CRR: 1.41; 95% CI: 1.10–1.77), diseases of the respiratory system (CRR: 1.25; 95% CI: 1.09–1.30), diseases of the circulatory system (CRR: 1.19; 95% CI: 1.12–1.26) and endocrine diseases (CRR: 1.13; 95% CI: 1.02–1.25). The majority of circulatory diseases are affected by extremely high temperature, among which sequelae of cerebrovascular diseases (CRR: 1.61; 95% CI: 1.41–1.85) had the highest CRR, followed by stroke (CRR: 1.52; 95% CI: 1.12–2.09), hypertension (CRR: 1.46; 95% CI: 1.26–1.69), cerebral infarction (CRR: 1.20; 95% CI: 1.08–1.35), ischemic heart disease (CRR: 1.18; 95% CI: 1.07–1.31) and intracerebral hemorrhage (CRR: 1.08; 95% CI: 1.00–1.16). Both acute infectious respiratory diseases and chronic respiratory diseases are sensitive to extreme heat, among which influenza and pneumonia had CRR of 1.28 (95% CI: 1.15–1.43), and chronic obstructive pulmonary disease had CRR of 1.22 (95% CI: 1.12–1.32). For extreme low temperature, the highest CRR of 1.46 (95% CI: 1.16–1.82) was observed for diseases of the circulatory system, followed by diseases of the respiratory system (CRR: 1.34; 95% CI: 1.07–1.42) and diseases of the digestive system (CRR: 1.23; 95% CI: 1.04–1.45). Although most circulatory diseases are susceptible only to extreme heat, ischemic heart disease and intracerebral hemorrhage are susceptible to both extreme cold and extreme heat, with CRR of 1.75 (95% CI: 1.26–2.44) and 1.49 (95% CI: 1.13–1.96), respectively. 

Table 3 also shows regional differences in the association between extreme temperature and mortality for a spectrum of diseases. For extreme heat effects, people living in subtropical zones are more sensitive to extreme heat than people living in temperate zones, and the difference was statistically significant (*p* = 0.023). Intracerebral hemorrhage (CRR: 1.12; 95% CI: 1.00–1.25) and diseases of the nervous system (CRR: 1.51 (1.04–2.19) are only sensitive in subtropical zones; ischemic heart disease (CRR: 1.28; 95% CI: 1.07–1.53), chronic obstructive pulmonary disease (CRR: 1.26; 95% CI: 1.09–1.45), influenza and pneumonia (CRR: 1.28; 95% CI: 1.14–1.44) and injuries (CRR: 1.34; 95% CI: 1.16–1.56) are sensitive in both subtropical and temperate zones, and the difference was not statistically significant (*p >* 0.05). For extreme cold temperature effects, intracerebral hemorrhage is sensitive in both subtropical and temperate zones, and the difference was not statistically significant (*p* = 0.3). Ischemic heart disease (CRR: 1.75; 95% CI: 1.32–2.32) and chronic obstructive pulmonary disease (CRR: 1.49; 95% CI: 1.30–1.71) are sensitive only in subtropical zones.

### 3.3. Attributable Fractions of Non-Optimum Temperatures

Table 4 shows the AFs of cause-specific mortality due to non-optimum temperatures in two regions in China. We found that non-optimum temperatures increased the mortality burden of the population, and the overall AF in low temperatures (9.40%; 95% eCI: 2.92–15.83%) was greater than in high temperatures (1.62%; 95% eCI: 0.76–2.43%). The highest AF of heat was observed for injuries (6.5%; 95% eCI: 2.5–10.0%), followed by nervous system diseases (4.2%; 95% eCI: 1.0–7.03%), circulatory system diseases (3.9%; 95% eCI: 0.42–7.09%), respiratory system diseases (1.85%; 95% eCI: 0.68–2.85%) and endocrine diseases (0.90%; 95% eCI: −0.3–1.9%). Circulatory system, respiratory system, and nervous system diseases had a greater burden attributed to heat in the subtropical zone, while injuries had a greater burden in the temperate zone. The highest AF of cold temperature was observed for respiratory system diseases (13.3%; 95% eCI: 5.6–23.6%), followed by circulatory system diseases (11.8%; 95% eCI: 2.4–19.6%). Circulatory system and respiratory system diseases had a greater burden attributed to cold temperature in the subtropical zone.

## 4. Discussion

Our study covered 17 study sites and 1,380,713 records of all-cause deaths in China. We systematically screened overall and regional heat- and cold-sensitive diseases from 18 specific diseases of eight disease systems according to the International Classification of Diseases (ICD-10) and assessed the mortality burden of sensitive diseases. We found that more diseases are sensitive to extreme heat than to extreme cold, but the mortality burden of the total population ascribed to low temperature is larger. To the best of our knowledge, this is the first study to systematically and comprehensively screen regional sensitive diseases associated with both extreme heat and cold and assess the mortality burden of sensitive diseases attributable to extreme non-optimum temperatures, especially in China.

Our findings show that extreme temperature can increase mortality due to multiple diseases, especially those that receive less attention such as nervous system diseases and injuries. The association between extreme non-optimum temperature and mortality from circulatory system [14,27,28], respiratory system [29,30,31] and endocrine [19,32,33,34] diseases has been widely reported. However, no study has assessed the association between extreme temperature and mortality due to nervous system diseases and injuries. Our findings show that extreme heat could increase mortality due to injury, and cold has no significant effect. Several experimental studies have shown that drivers who experience fatigue in high-temperature environments tend to have more technical errors and are more likely to deviate from the lane, increasing the occurrence of vehicle accidents [35,36,37]. In many sites located in the subtropics in our study, high temperature tends to be accompanied by greater chances of rain, which can increase the incidence of vehicle accidents [38]. What is more, studies have demonstrated that intense and prolonged exposure to extreme temperature is associated with health effects, such as dehydration, spasms and fatigue, which can increase the incidence of accidental injuries [16,17,39]. Our study found that extreme heat can increase mortality due to nervous system diseases. Both experimental and epidemiological studies [40] have implied that extreme heat can affect the immune system, which plays an important role in the pathogenesis and progression of nervous system diseases such as Parkinson’s disease [41]. Therefore, it was speculated that extreme heat could affect the nervous system. 

In addition, we further explored regional differences in the association between extreme temperature and mortality. Our findings show that people living in subtropical zones are more sensitive to extreme temperature. This is inconsistent with previous studies [18,42]. A possible reason is that the division of the study area or the health outcomes was different. For example, Ma et al. [42] divided study regions into northern, eastern and southern areas to explore the regional differences in the impact of heat waves on mortality, and the results showed that people living in northern regions were more sensitive to heat. Zhao et al. [18] divided study regions into northern, central and southern areas by latitude to explore the regional differences in the influence of extreme temperature on emergency department visits, and the results showed that people living in the northern areas were more sensitive to heat. In any case, our findings provide a scientific basis for identifying diseases sensitive to extreme temperature in different climate zones, and suggest that it is necessary to consider local climate characteristics and geographic location when healthcare providers and public health authorities are developing response plans to protect vulnerable groups from extreme temperature [43]. However, in this study, only 17 sites were included to explore regional differences. Further investigation is needed for multi-city and multi-regional research.

We also evaluated the mortality burden attributable to non-optimum temperatures for all 17 study sites. Our findings show that 11.03% of all-cause mortality could be attributed to non-optimum temperatures, which was comparable to China’s estimate of 11.00% reported in the global analysis [3] and 14.33% reported in 272 Chinese cities [12]. The mortality burden attributable to non-optimum temperatures was different by climate zones. Cities in subtropical zones have no central heating in the winter and high temperature is frequent in the summer, leading to a heavy mortality burden of effects [44]. It was indicated that people who have more chances to be exposed to extreme temperatures are probably more vulnerable to extreme temperature conditions [42,45].

Several limitations should be acknowledged. First, the daily mortality numbers for several diseases, such as stroke, infectious diseases and nervous system diseases were small, which might underestimate or overestimate the exposure–response relationship. Also, the small number of study sites with intra-regional variation in weather and air pollution might have relatively poor representativeness for regional risk analysis. Second, as with most previous epidemiological studies, we used temperature data from outdoor monitors at a fixed location, rather than individual direct measurements, which could lead to exposure to measurement errors. Third, the uncertainty of individual behavior was not taken into account in our study. For example, during times of extreme temperature, people often choose to stay indoors or take preventive measures, which might underestimate the relative risk. However, our results warrant further research on climate-sensitive disease screening and health risk analysis with more sites and more death cases, especially for rare diseases. 

## 5. Conclusions

Our study comprehensively and systematically screened regional diseases sensitive to extreme temperature and assessed regional cause-specific mortality burden ascribed to extreme temperature. Both extreme heat and cold temperature can increase mortality in multiple regions of China, but the strength of association varies by cause of death and region. Therefore, it is important to understand a spectrum of diseases sensitive to extreme temperatures in different regions, which would provide evidence to warrant taking region-specific preventive measures to reduce the mortality burden in China, particularly in the context of rapid climate change.

## Figures and Tables

**Figure 1 ijerph-17-00184-f001:**
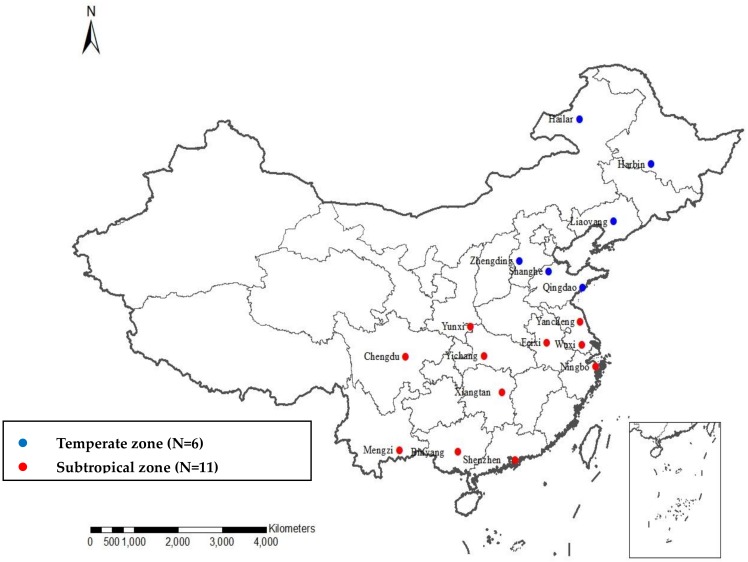
Geographical location of 17 study sites.

**Figure 2 ijerph-17-00184-f002:**
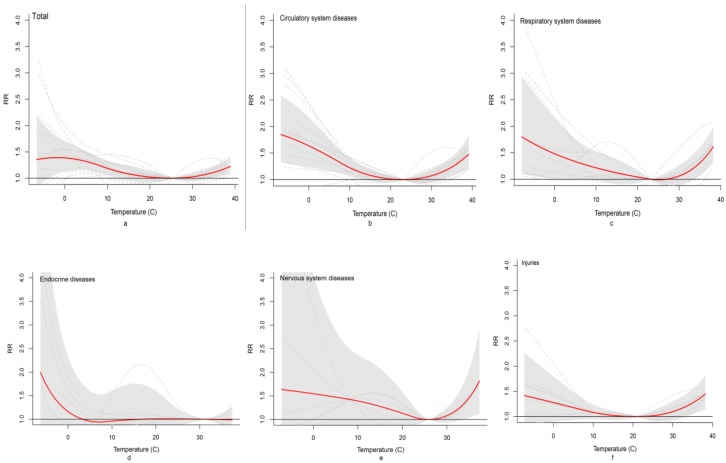
Overall exposure–response relationship between daily maximum temperature and cause-specific mortality by 30 d lag at 17 study sites in China: (**a**–**f**) total, circulatory system diseases, respiratory system diseases, endocrine diseases, nervous system diseases, and injuries, respectively.

**Table 1 ijerph-17-00184-t001:** Descriptive statistics of meteorological factors and air pollutants at 17 study sites in China, 2014–2017. MMT, minimum mortality temperature.

Study Site	Study Period	Population	Maximum Temperature (°C)						Mean Relative Humidity (%)			Mean Barometric Pressure (hpa)			Mean PM2.5 (μg/m^3^)			Mean O_3_ (μg/m^3^)		
(city/county)		(million)	Mean ± SD	Min	Max	MMT	97.5th	2.5th	Mean ± SD	Min	Max	Mean ± SD	Min	Max	Mean ± SD	Min	Max	Mean ± SD	Min	Max
Harbin	2014–2016	31.7	10 ± 15.2	–21.6	36.2	26	32	–16	65 ± 15	15	97	9994 ± 95	9732	10,252	64.8 ± 61	8	653	63.6 ± 42.7	10	179
Liaoyang	2014–2015	1.89	16.1 ± 12.7	–17.1	37.1	29.5	33	–7	56 ± 16	13	98	10,123 ± 96	9840	10,359	33 ± 28	3	423	82.3 ± 32.1	17	291
Hailar	2014–2017	0.29	5.8 ± 17.9	–34.1	41.7	24	32	–24	62 ± 15	15	94	9371 ± 76	9127	9618	29.1 ± 17	5	164	74.4 ± 72.4	13	160
Zhengding	2014–2016	2.94	20.5 ± 11.1	–9.1	43.4	18	37	1	54 ± 20	12	99	10,076 ± 99	9868	10,346	108 ± 89	0	653	102 ± 67	0	322
Qingdao	2014–2016	9.39	17.3 ± 9.3	–7.7	36.9	29	32	1	69 ± 16	16	100	10,075 ± 90	9878	10,292	48 ± 34	4	298	102 ± 45	17	277
Shanghe	2014–2016	1.87	19.9 ± 10.6	–10.6	40	17	35	1	68 ± 16	23	100	10,145 ± 99	9949	10,416	79 ± 52	8	342	122 ± 48	5	314
Wuxi	2014–2016	19.6	21.4 ± 9	–3.8	40.6	25	36	6	75 ± 13	33	100	10,153 ± 92	9945	10,410	61 ± 26	11	223	103 ± 48	10	279
Yancheng	2014–2017	8.25	20.2 ± 9.1	–6.2	39	28	35	4	76 ± 13	34	100	10,159 ± 92	9945	10,406	49 ± 35	5	226	83 ± 48	3	262
Feixi	2014–2017	4.00	21.5 ± 9.1	–3.2	40.8	20.5	37	5	77 ± 12	32	99	10,133 ± 102	8586	10,424	51.8 ± 35.6	3	372	54 ± 43	12	251
Yichang	2014–2017	3.67	20.9 ± 8.8	0.4	38.3	30	36	5	76 ± 14	46	58	9852 ± 85	9692	10,126	71 ± 41	6	343	73 ± 42	10	198
Yunxi	2014–2016	1.59	21.7 ± 9.4	–0.4	41.5	30	38	5	73 ± 14	14	99	9828 ± 95	9621	10,084	45 ± 33	0	554	83 ± 48	0	183
Chengdu	2013–2017	16.3	21.5 ± 7.8	2.8	36.7	30.5	35	8	80 ± 9	42	98	9506 ± 74	9325	9770	70 ± 49	9	423	89 ± 49	7	278
Ningbo	2014–2016	8.20	21.9 ± 8.6	–2.3	39.2	30	36	6	80 ± 11	34	100	10,153 ± 88	9857	10,397	43 ± 26	7	202	94 ± 49	6	242
Xiangtan	2014–2016	2.85	23.4 ± 8.8	0.1	40	20	37	5	82 ± 12	38	100	10,071 ± 88	9911	10,368	51 ± 33	0	236	81 ± 48	0	279
Mengzi	2014–2017	1.62	25.1 ± 5.5	1.3	35.4	24	32	13	69 ± 12	26	100	8677 ± 40	8580	8813	19 ± 31	1	61	84 ± 40	12	175
Shenzhen	2016–2017	24.4	27.1 ± 5.6	6.5	36.9	25.5	34	14	75 ± 13	19	100	10,029 ± 64	9765	10,223	30 ± 17	6	110	82 ± 49	25	244
Binyang	2014–2016	3.26	25.9 ± 7.3	6.5	37.3	31.5	35	10	80 ± 11	36	100	9975 ± 74	9784	10,228	29 ± 19	4-	117	89 ± 34	24	196

**Table 2 ijerph-17-00184-t002:** Summary of descriptive statistics on average daily cause-specific mortality at 17 study sites in China, 2014–2017.

Variables	Mean ± SD	Minimum	Maximum
Total	63 ± 66	4	222
Diseases of circulatory system	21 ± 24	2	92
Hypertension	1.2 ± 1.6	0	6
Ischemic heart disease	8 ± 11	0	45
Cerebrovascular disease	11 ± 13	0	41
Cerebral infarction	3 ± 5	0	17
Intracerebral hemorrhage	4 ± 5	0	18
Stroke	1 ± 1	0	3
Sequelae of cerebrovascular disease	2 ± 3	0	8
Diseases of respiratory system	8 ± 13	0	53
Chronic lower respiratory disease	6 ± 10	0	40
Influenza and pneumonia	2 ± 3	0	10
Diseases of digestive system	1 ± 2	0	7
Diseases of genitourinary system	1 ± 1	0	2
Endocrine diseases	2 ± 2	0	6
Diabetes	2 ± 2	0	6
Diseases of nervous system	1 ± 1	0	3
Infectious diseases	1 ± 1	0	3
Injuries	4 ± 4	0	13

Note: Total means all-cause deaths.

**Table 3 ijerph-17-00184-t003:** Cumulative relative risks of cause-specific mortality due to extreme heat and cold in two regions in China.

Region	Extreme Heat			Extreme C*o*ld		
	Overall	Subtropical Zone	Temperate Zone	Overall	Subtropical Zone	Temperate Zone
**Total**	**1.13 (1.09, 1.18)**	**1.18 (1.08, 1.25)**	**1.06 (1.02, 1.15)**	**1.30 (1.10, 1.54)**	**1.34 (1.11, 1.62)**	1.14 (0.98, 1.32)
**Circulatory system**	**1.19 (1.12, 1.26)**	**1.27 (1.13, 1.43)**	**1.14 (1.03, 1.25)**	**1.46 (1.16, 1.82)**	**1.54 (1.20, 1.97)**	1.47 (0.85, 2.54)
Hypertension	**1.46 (1.26, 1.69)**	—	—	1.64 (0.91, 2.93)	—	—
Ischemic heart disease	**1.18 (1.07, 1.31)**	**1.28 (1.07, 1.53)**	**1.10 (1.00, 1.21)**	**1.75 (1.26, 2.44)**	**1.75 (1.32, 2.32)**	2.16 (0.82, 5.67)
Cerebrovascular disease	**1.19 (1.12, 1.26)**	**1.23 (1.14, 1.33)**	**1.08 (1.01, 1.16)**	**1.39 (1.09, 1.76)**	**1.71 (1.43, 2.03)**	0.78 (0.56, 1.08)
Cerebral infarction	**1.20 (1.08, 1.35)**	—	—	1.49 (0.99, 2.26)	—	—
Intracerebral hemorrhage	**1.08 (1.00, 1.16)**	**1.12 (1.00, 1.25)**	0.96 (0.86, 1.07)	**1.49 (1.13, 1.96)**	**1.32 (1.13, 1.99)**	**1.45 (1.15, 1.90)**
Stroke	**1.52 (1.12, 2.09)**	—	—	1.29 (0.57, 2.93)	—	—
Sequelae of cerebrovascular disease	**1.61 (1.41, 1.85)**	—	—	1.18 (0.75, 1.85)	—	—
**Respiratory system**	**1.25 (1.09, 1.30)**	**1.30 (1.17, 1.45)**	1.18 (0.96, 1.45)	**1.34 (1.07, 1.42)**	**1.33 (1.14, 1.48)**	0.90 (0.68, 1.43)
Influenza and pneumonia	**1.28 (1.15, 1.43)**	**1.28 (1.14, 1.44)**	**1.29 (1.04, 1.61)**	1.36 (0.76, 2.41)	0.92 (0.70, 1.17)	1.20 (0.74, 1.96)
Chronic obstructive pulmonary disease	**1.22 (1.12, 1.32)**	**1.26 (1.09, 1.45)**	**1.41 (1.07, 1.85)**	**1.35 (1.10, 1.65)**	**1.49 (1.30, 1.71)**	0.94 (0.76, 1.17)
**Endocrine diseases**	**1.13 (1.02, 1.25)**	—	—	0.94 (0.51, 1.73)	—	—
Diabetes	**1.13 (1.03, 1.25)**	—	—	0.95 (0.50, 1.81)	—	—
**Digestive system**	1.22 (0.95, 1.56)	—	—	**1.23 (1.04, 1.45)**	—	—
**Genitourinary system**	1.03 (0.55, 1.92)	—	—	1.13 (0.35, 3.63)	—	—
**Nervous system**	**1.41 (1.10, 1.77)**	**1.51 (1.04, 2.19)**	1.28 (0.88, 1.87)	1.28 (0.64, 3.43)	1.92 (0.87, 4.22)	0.36 (0.06, 2.00)
**Injuries**	**1.45 (1.27, 1.65)**	**1.34 (1.16, 1.56)**	**1.27 (1.15, 1.36)**	1.17 (0.95, 1.44)	1.21 (0.98, 1.51)	1.49 (0.83, 2.65)
**Infectious diseases**	0.43 (0.07, 2.58)	—	—	1.00 (0.2, 4.96)	—	—

Note: “—” indicates no calculation of cumulative relative risk (CRR) due to few daily mortality data. Extreme heat and cold: 97.5th and 2.5th percentiles of daily maximum temperature distribution, respectively. Bold data represent statistical significance, and bold fonts represent 8 major systems.

**Table 4 ijerph-17-00184-t004:** Attributable fractions (%) of cause-specific mortality due to high and low temperatures in two regions in China.

Cause of Death	AF_all_	High Temperature			Low Temperature		
		Overall	Subtropical Zone	Temperature Zone	Overall	Subtropical Zone	Temperature Zone
Total	**11.03**	**1.62 (0.76, 2.43)**	**1.96 (0.95. 2.91)**	**0.99 (0.42, 1.56)**	**9.40 (2.92, 15.83)**	**9.06 (4.67, 13.53)**	10.0 (−1.8, 20.1)
Circulatory system	**15.7**	**3.9 (0.42, 7.09)**	**2.5 (0.8, 4.1)**	6.2 (−0.13, 12.0)	**11.8 (2.4, 19.6)**	**15.1 (4.1, 24.6)**	6.5 (−0.4, 11.5)
Respiratory system	**15.2**	**1.85 (0.68, 2.85)**	**2.6 (1.3, 3.8)**	0.58 (−0.26, 1.26)	**13.3 (5.6, 23.6)**	**24.2 (9.8, 36)**	−4.3 (−15.8, 3.54)
Endocrine	9.3	0.9 (−0.3, 1.9)	—	—	8.4 (−15.6, 25.0)	—	—
Nervous system	12.6	**4.2 (1.0, 7.03)**	**5.03 (1.8, 7.8)**	1.8 (−;1.6, 4.4)	8.4 (−21.1, 32.4)	16.0 (−5.1, 29.2)	−16.5 (−48.2, 56.7)
Injuries	10.1	**6.5 (2.5, 10.0)**	**6.2 (2.2, 9.8)**	**7.6 (3.5, 31.1)**	3.6 (−10.6, 14.1)	3.8 (−7.1, 12.0)	2.9 (−21.5, 20.5)

Note: “—” indicates no calculation of attributable fractions (AFs) due to few daily mortality data. High temperature means MMT to maximum daily maximum temperature. Low temperature means minimum daily maximum temperature to MMT. AF_all_ represents total attributable fractions ascribed to high and low temperatures. Bold data represent statistical significance, and bold fonts represent 8 major systems.

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
