# Peer review of "Regional Temperature-Sensitive Diseases and Attributable Fractions in China"

_ijerph, 2019, doi:10.3390/ijerph17010184_

Round 1
Reviewer 1 Report
Sir,
thank you for the opportunity to review this extensively revised version of a paper I still think would deserve a full publication on IJERPH.
I've no further methodological concerns.
Still, Authors should be aware that minor, annoying typos remains annoyingly scattered across the text. Some examples:
"In addition, Foreign studies [17,18]such as Australia and the United Kingdom have mentioned that high-temperature environments can increase accidents such as vehicle accidents and occupational injuries, but such research is still lacking in China".
Did you mean:
"In addition, some studies performed in Australia and UK have mentioned that high-temperature environments can increase accident rates, both as vehicle and occupational injuries..."
?
"That is to say people live in high latitude may be more vulnerable to high temperature"
did you mean:
"In other words, people living in high latitudes..."
?
Similarly, please revise the "limitation" subsection of the discussion.
Finally, as a consequence of the model you assessed, I warmly recommend to include in the discussion the following study, that includes a model very similar to that you employed, focusing on Occupational settings.
https://pubmed.ncbi.nlm.nih.gov/31427308-do-exposure-to-outdoor-temperatures-no-2-and-pm-10-affect-the-work-related-injuries-risk-a-case-crossover-study-in-three-italian-cities-2001-2010/?from_single_result=occupational+injuries+pollution+marinaccio
MR
Author Response
Dear Reviewer,
We greatly appreciate the time that you have spent and the constructive comments that have improved our manuscript. The manuscript has been revised in response to the comments and suggestions. Below are point-by-point responses to the comments and the corresponding changes in the manuscript (all changes in the text have been highlighted). Please see the attachment and manuscript for details.
Kind regards.
Xuemei Su

Reviewer 2 Report
I think the authors made an effort and the manuscript improved respect the original submission.
There are sections that are still unclear, in particular it is not clear how the CRR are derived. From the statistical analysis section (lines 124-128) it looks that the CRR are derived from the pooled curve obtained using the multivariate meta-regression model. In the paragraph above (lines 119-121) it looks that the CRR were derived from first-stage curves and then meta-analysed. The author's response on this point suggests that this second approach was used, but this need to be clarified in the manuscript. Note that the multivariate meta-regression method cited as (16) was introduced in Gasparrini 2012 (Stat med) and Gasparrini 2013 (BMCMRM).
Figure 2 looks containing duplicate graphs for each diseases. For example the second graph for total mortality simply add first stage-curves to the pooled curve already represented in the first graph.
I would say that Figure 2 shows the temperature-mortality association and not just the MMT as indicated in line 206.
Perhaps the authors mean (p>0.05) in line 235 as the difference was not significative.
Line 263. I think that diseases of the circulatory system doesn't have an higher heta-AF% in subtropical zones.
Is the sentence in lines 267-268 supported by inferential procedures?
Author Response

(The authors gave the same response as above.)

Reviewer 3 Report
The authors have addressed most of my previous comments. Although I would still argue that using two equally spaced internal knots for temperature (an early setting for DLNM) is arbitrary and would prefer the authors to use more recent settings of the DLNM (e.g., three internal knots using natural cubic splines with 4 df), the current revision is fine.
Author Response
Dear Reviewer,
We greatly appreciate the time that you have spent and the constructive comments that have improved our manuscript. The manuscript has been revised in response to the comments and suggestions. Below are point-by-point responses to the comments and the corresponding changes in the manuscript (all changes in the text have been highlighted).
Point 1: The authors have addressed most of my previous comments. Although I would still argue that using two equally spaced internal knots for temperature (an early setting for DLNM) is arbitrary and would prefer the authors to use more recent settings of the DLNM (e.g., three internal knots using natural cubic splines with 4 df), the current revision is fine.
Repose 1: Thank you very much for your detailed and rigorous comments. Based on your first two rounds of revision comments, I've learned a lot through constant exploration.
Kind regards.
Xuemei Su

This manuscript is a resubmission of an earlier submission. The following is a list of the peer review reports and author responses from that submission.
Round 1
Reviewer 1 Report
Sir,
First at all, thank you for the opportunity ti review this significant research paper.
In my opinion, not only this Is a very well written article, but its potential significance for public health urges for a Quick publication.
Only minor recommendations that, alas!, must be fixed before publication.
1) the title includes an annoying misspelling (reginal for regional), please fix it
2) table 3 has some formatting issues at the row "stroke", and It should be similarly fixed
3) introduction and discussion don't include references on heat related injuries, even though a solid base of evidence has increasingly shed light on the role of temperatures: I suggest the Authors to include some references and discuss them, particularly as the theme of u-curve vs. J-curve vs. Linear association of temperatures with events has been extensively addressed in such researches, including some potential explanations (e.g. avoiding exposures and increased referral of preventive measures in extremes temperatures) that may contribute to their discussion (for example but not limited to:
https://www.ncbi.nlm.nih.gov/pubmed/30675732
https://www.ncbi.nlm.nih.gov/pubmed/29165429
https://www.ncbi.nlm.nih.gov/pubmed/31412493
Author Response
Dear reviewer:
I am very grateful to your comments for the manuscript. According with your advice, we amended the relevant part in manuscript. Please see the attachment.
Sincerely,
Xuemei Su

Reviewer 2 Report
This study analysed the temperature-mortality association in 17 Chinese cities. The association is evaluated considering overall and cause-specific deaths as outcome. In particular the authors consider as outcomes deaths classified using ICD10 codes for nervous system, infectious diseases and injury deaths. The author also attempt to characterise geographically the associations between the temperature and the different outcomes.
Overall the study is well conducted with statistical methods coherent with the study design.
The main problem of this study is the possible low power on exanimating as possible outcomes nervous system, infectious diseases and injury deaths. For all these conditions there are less than 10,000 deaths in the observation period. This mean that the study have a low power to detect associations for these condition with a high probability to report false positive results. This is particularly true for the sub-analysis by climatic regions.
A second main issue is the analytic method the author choose to evaluate differences among the two regions. My understanding is that the author performed a subgroup analysis pooling the city specific estimates on each of the two sub-regions. I think that the differences across regions should be tested in a meta-regression model. This procedure would ensure more robust results with the possibility to use the all sample on calculating the p-values to support the statement of differences among regions.
There is some less of clarity in the statistical analysis section that doesn’t allow to fully understand the results:
1. It is not clear if the meta-analysis was performed considering the reduced spline-coefficients in a multivariate meta-analytic model as suggested in Figure 2, or the meta-analysis was performed on the summary indices (e.g. 97.5Th vs MMT) using univariate models. The results showed in Figure 2 does not align with RRs presented in Table 3; for example the RR for endocrine disease is 1.19 (1.09; 1.31) in table 3, but confidence bands cross the Null Hypothesis value 1.0 in Figure 3. Moreover the RRs for disease of the nervous system and injury looks higher than CVD or respiratory disease in Table 3, but not in Figure 2.
2. The AF% reported in Table 4 does not align with RRs in table 3, for example for endocrine disease the heat AF% changes dramatically by regions in table 4, while the RRs are pretty stable in table 3.
In descriptive tables an empty space should precede and follow the sign “±”.
The title have an error: “Reginal” instead of “Regional”.
Author Response

(The authors gave the same response as above.)

Reviewer 3 Report
This manuscript attempts to systematically screen regional temperature-sensitive diseases and to assess the attributable burden of cause-specific mortality in 17 Chinese cities. Among the eight disease systems, the authors found a significant heat-related burden in 5 disease systems, including the nervous disease, injury, circulatory, respiratory, and endocrine system diseases, whereas only three disease systems (respiratory, circulatory disorders and injury) had significant cold-attributable burdens. Some regional differences in attributable burden estimates were also reported between the temperate monsoon zone and the subtropical monsoon zone. Overall, this paper addresses an important topic expanding the temperature-related health impacts on understudied disease systems. However, there are several issues that I believe should be addressed before getting published in this journal.
Major comments
The ICD-10 codes for certain diseases warrant further clarification. Why using K00-K99 for the digestive system, although it seems the WHO 2016-version of ICD-10 doesn’t have K95-K99 in the coding? The term “stroke” for I64 only refers to unspecified stroke, which doesn’t include ischemic or hemorrhagic stroke. For the “cross-basis” function of temperature, it is not clear how it was constructed, e.g., what is the spline function and knots/df used for the exposure and lag dimensions? Why use a maximum of 30 days for the lag structure? Sensitivity analyses using 7, 14, and 21 days should be conducted. A few questions for the adjustment of air pollution:
1) why use natural splines for air pollutants? The majority of air pollution time-series studies showed that the exposure-response relationship for PM2.5, CO, and O3 are all linear;
2) How about the lag structure for air pollutants? The current day is not generally the lag that presents the largest effect for air pollutants;
3) What are the correlations among the air pollutants? Why in the main model adjusted all three pollutants?
A major issue in calculating the attributable fraction for each city is that the city-specific exposure-response (ER) curves are bit messy (as shown in Figure 2). So typically, in the second stage, one needs to use BLUP (Best Linear Unbiased Predictions) to get the adjusted city-specific ER curves, which borrow information of the overall ER curve. Since all the attributable burden analyses were calculated based on the city-level estimates, this additional adjustment using BLUP may help improve the precision of estimates. The focus of this paper seems inconsistent regarding the temperature ranges. Moderate temperatures were used but their results were never presented. In Table 3. Why only showing the RR for extreme temperatures? What are the results of moderate temperatures? However, in Table 4, results were shown for the hot (extreme + moderate) and cold (extreme + moderate). Given previous studies generally found a much larger death burden to moderate rather than the extreme temperatures, this might be interesting. Lines 240-249 only discussed the potential mechanism for heat, what about the cold? What’s the plausibility of cold-related injury? The regional difference in temperature-related disease burden, which was inconsistent with most of the previous studies, seems to be biased due to the smaller number of cities (6 vs. 11) and smaller population sizes for the northern cities in the temperature monsoon zone vs. southern cities in the subtropical monsoon zone.
Minor comments
Line 73-74, based on Figure 1, the number of cities in the subtropical zone should be 11. Line 194, where are the results for “tumor”? The analysis of the tumor was never mentioned before. Figure 2. Why only present significant relationships? How about other disease systems? The nonsignificant results should also be presented, maybe in the supplemental figure.
4.Table S1. The format needs adjustment. For example, it is not clear for Chengdu, the “1.26(1.12,1.42)” belongs to the “extract CO” column or the one with 5 df for relative humidity/barometric pressor.
Author Response

(The authors gave the same response as above.)

Round 2
Reviewer 2 Report
The authors have positively answered to the issues I rose in my first review.
Perhaps they could have used a different analytic approach, deriving the CRR from the meta-regression model using 21 days lag. This would avoid some incongruences between temperature-mortality associations curves, CRR and AF.
I still think that regional differences need to be assessed through inferential procedures and not comparing significances of CRR in different regions. This is an old way to assess interactions and it's highly discouraged.
Author Response
Dear reviewer;
Thank you very much for carefully reviewing my manuscript.Based on your review suggestions, I have made detailed changes. Please see the attachment for details.
Kind regards.
Yours sincerely,
Xuemei Su

Reviewer 3 Report
I appreciate the authors' effort in addressing most of my comments. However, several issues are critical to the robustness of the findings and should be addressed.
The lag structure of air pollutants matter as demonstrated in the vast body of evidence in air pollution epidemiology. Insufficient control of air pollution (e.g., the current day might not be the largest and significant lag structure for a specific air pollutant) may lead to bias in the temperature effect estimates. This issue cannot simply be disregarded by saying “we would like to consider the lag structure for air pollutants in future research.”
Similarly, Figure 2 demonstrates a massive difference between the city-specific exposure-response (ER) curves and the pooled curve. Since the authors are using Distributed Lag Nonlinear Model (DLNM) and using BLUP is the standard way to get the city-level estimates for temperature studies using DLNM, I don’t see any reasonable explanation of not using BLUP in this analysis. Note that I am referring to the overall shape of the ER curves, not the MMT for each city.
Regarding the crossbasis function for temperature, one concern is the equalknots() used for temperature term. The current setting [equalknots(lndn$tmax, fun=”bs,” df=4, degree=2] only arbitrarily put two equalknots for temperature, which is different than what is usually done in the temperature studies using DLNM. Sensitivity analysis on the parameters for the temperature exposure dimension in DLNM should be done, such as using three internal knots at the 10th, 75th, and 90th percentiles of temperature or using natural cubic splines with four df.
Author Response
Dear reviewer:
Thank you very much for reviewing my manuscript. Based on your review suggestions, I have made detailed changes. See the attachment for details.
Kind regards.
Yours sincerely,
Xuemei Su
